# Azacitidine vs. Decitabine in Unfit Newly Diagnosed Acute Myeloid Leukemia Patients: Results from the PETHEMA Registry

**DOI:** 10.3390/cancers14092342

**Published:** 2022-05-09

**Authors:** Jorge Labrador, David Martínez-Cuadrón, Adolfo de la Fuente, Rebeca Rodríguez-Veiga, Josefina Serrano, Mar Tormo, Eduardo Rodriguez-Arboli, Fernando Ramos, Teresa Bernal, María López-Pavía, Fernanda Trigo, María Pilar Martínez-Sánchez, Juan-Ignacio Rodríguez-Gutiérrez, Carlos Rodríguez-Medina, Cristina Gil, Daniel García Belmonte, Susana Vives, María-Ángeles Foncillas, Manuel Pérez-Encinas, Andrés Novo, Isabel Recio, Gabriela Rodríguez-Macías, Juan Miguel Bergua, Víctor Noriega, Esperanza Lavilla, Alicia Roldán-Pérez, Miguel A. Sanz, Pau Montesinos

**Affiliations:** 1Hematology Department, Research Unit, Complejo Asistencial Universitario de Burgos, 09006 Burgos, Spain; 2Hematology Department, Hospital Universitari i Politécnic La Fe, 46026 Valencia, Spain; martinez_davcua@gva.es (D.M.-C.); rodriguez_reb@gva.es (R.R.-V.); msanz@uv.es (M.A.S.); 3Hematology Department, MD Anderson Cancer Center Madrid, 28033 Madrid, Spain; afuente@mdanderson.es; 4Hematology Department, Hospital Universitario Reina Sofía, IMIBIC, 14004 Córdoba, Spain; josefina.serrano.sspa@juntadeandalucia.es; 5Hematology Department, Hospital Clínico Universitario de Valencia, Instituto de Investigación Sanitaria—INCLIVA, 46010 Valencia, Spain; tormo_mar@gva.es; 6Hematology Department, Hospital Universitario Virgen del Rocío, Instituto de Biomedicina de Sevilla (IBIS/CSIC), 41013 Sevilla, Spain; eduardo.rodriguez.arboli.sspa@juntadeandalucia.es; 7Hematology Department, Hospital Universitario de León, 24071 León, Spain; framoso@saludcastillayleon.es; 8Hematology Department, Hospital Universitario Central Asturias, ISPA, IUOPA, 33011 Oviedo, Spain; bernaldelcastillo@gmail.com; 9Hematology Department, Hospital General de Valencia, 46026 Valencia, Spain; lopez_marpav@gva.es; 10Hematology Department, Centro Hospitalar Universitário de São João, 4200-319 Porto, Portugal; ftrigo@chsj.min-saude.pt; 11Hematology Department, Hospital Universitario 12 de Octubre, 28041 Madrid, Spain; mariapilar.martinez.sanchez@salud.madrid.org; 12Hematology Department, Hospital Universitario Basurto, 48013 Bilbao, Spain; juanignacio.rodriguezgutierrez@osakidetza.eus; 13Hematology Department, Hospital Universitario de Gran Canaria Doctor Negrín, 35010 Las Palmas de Gran Canaria, Spain; crodmedk@gobiernodecanarias.org; 14Hematology Department, Hospital General Universitario de Alicante, 03010 Alicante, Spain; gil_cricor@gva.es; 15Hematology Department, Hospital Universitario Sanitas La Zarzuela, 28023 Madrid, Spain; dgarciabe@sanitas.es; 16Hematology Department, Hospital Germans Trias i Pujol-ICO, Josep Carreras Research Institute, Universitat Autònoma de Barcelona, 08916 Badalona, Spain; svives@iconcologia.net; 17Hematology Department, Hospital Universitario Infanta Leonor, 28031 Madrid, Spain; mariaangeles.foncillas@salud.madrid.org; 18Hematology Department, Hospital Clínico Universitario de Santiago de Compostela, 15706 Santiago de Compostela, Spain; manuel.mateo.perez.encinas@sergas.es; 19Hematology Department, Hospital Universitario Son Espases, 07120 Palma de Mallorca, Spain; andres.novo@ssib.es; 20Hematology Department, Complejo Asistencial de Ávila, 05071 Avila, Spain; irecio@saludcastillayleon.es; 21Hematology Department, Hospital General Universitario Gregorio Marañón, 28007 Madrid, Spain; gabriela.rodriguez@salud.madrid.org; 22Hematology Department, Hospital San Pedro de Alcántara, 10003 Caceres, Spain; jmberguaburg@gmail.com; 23Hematology Department, Hospital Universitario de A Coruña, 15006 La Coruna, Spain; victor.noriega.concepcion@sergas.es; 24Hematology Department, Hospital Universitario Lucus Augusti, 27003 Lugo, Spain; esperanza.lavilla.rubira@sergas.es; 25Hematology Department, Hospital Universitario Infanta Sofía, 28702 San Sebastián de los Reyes, Spain; aroldanp@salud.madrid.org

**Keywords:** acute myeloid leukemia, elderly, treatment, azacitidine, decitabine, hypomethylating agents, PETHEMA

## Abstract

**Simple Summary:**

The use of azacitidine (AZA) and decitabine (DEC) have allowed more elderly acute myeloid leukemia (AML) patients to be treated. However, scarcely any direct comparative data exist between both drugs. This study shows no significant differences in response rates or overall survival (OS) between upfront AZA and DEC treatment in a large retrospective with long-term follow-up cohort of AML patients. However, we identified for the first time the baseline characteristics of patients benefitting from AZA vs. DEC in terms of responses, 120-day mortality and OS. We also show differences in salvage treatment patterns and outcomes after failure to both hypomethylating agents in a real-life setting. Taken together, these findings could help to select the most appropriate hypomethylating agent in monotherapy.

**Abstract:**

The hypomethylating agents, decitabine (DEC) and azacitidine (AZA), allowed more elderly acute myeloid leukemia (AML) patients to be treated. However, there are little direct comparative data on AZA and DEC. This multicenter retrospective study compared the outcomes of AZA and DEC in terms of response and overall survival (OS). Potential predictors associated with response and OS were also evaluated. A total of 626 AML patients were included (487 treated with AZA and 139 with DEC). Response rates were similar in both groups: CR was 18% with AZA vs. 23% with DEC (*p* = 0.20), CR/CRi was 20.5% vs. 25% (*p* = 0.27) and ORR was 32% vs. 39.5% (*p* = 0.12), respectively. Patients with leukocytes < 10 × 10^9^/L, bone marrow blasts < 50% and ECOG ≥ 2 had higher ORR with DEC than with AZA. OS was similar in both groups: 10.4 months (95% CI: 9.2–11.7) vs. 8.8 months (95% CI: 6.7–11.0, *p* = 0.455), for AZA and DEC, respectively. Age (≥80 years), leukocytes (≥ 10 × 10^9^/L), platelet count (<20 × 10^9^/L) and eGFR (≥45 mL/min/1.73 m^2^) were associated with higher OS with AZA compared to DEC. In conclusion, we found no differences in response and OS rates in AML patients treated with AZA or DEC.

## 1. Introduction

Acute myeloid leukemia is a hematopoietic neoplasm characterized by a clonal proliferation of abnormally differentiated myeloid precursors [1]. Treatment for older patients who are medically unfit for intensive chemotherapy (IC) remains a challenge. Treatment options include hypomethylating agents (HMAs), such as azacitidine (AZA) and decitabine (DEC), or low-dose cytarabine (LDAC) given alone or in combination with venetoclax [2,3,4].

Both HMAs were associated with a clinically meaningful improvement in overall survival (OS) compared to LDAC, although the primary end point was not met [5,6]. In the Phase 3 DACO-016 trial, DEC achieved a significantly higher median OS (7.7 months, 95% CI: 6.2–9.2) compared to patients receiving therapy choice, considered best supportive care (BSC) or LDAC (5.0 months, 95% CI: 4.3–6.3) in an ad hoc mature analysis in the intent-to-treat population [5]. Similarly, in the Phase 3 AML-001 trial, AZA patients exhibited a median OS (10.4 months, 95% CI: 8.0–12.7) compared to conventional care regimens, considered standard IC, LDAC or BSC (6.5 months, 95% CI: 5.0–8.6) [6]. However, there are no clinical trials comparing the efficacy of AZA vs. DEC, and there are only a few studies evaluating and comparing their outcomes [7,8,9,10]. To select the most appropriate HMA, as a backbone for combination or in monotherapy, could be relevant to optimize the management of unfit patients.

We aim to compare patients’ and disease characteristics, and the effectiveness of treatment with AZA or DEC in terms of response and OS in unfit newly diagnosed AML patients included in the Programa Español para el Tratamiento de las Hemopatías Malignas (PETHEMA) epidemiologic registry. We also aim to evaluate the potential predictors associated with response and OS with each drug.

## 2. Patients and Methods

### 2.1. PETHEMA Registry

The PETHEMA registry (NCT02607059) is an epidemiological registry of adult patients with newly diagnosed AML. It retrospectively collects the characteristics of AML and the usual practices of the Spanish centers, irrespective of the age of the patients and the treatment received.

### 2.2. Eligibility

Patients with an AML diagnosis (according to the WHO 2008 criteria [11,12], excluding acute promyelocytic leukemia) from 2006 until 2019, and treated upfront with HMA, were eligible for this retrospective analysis. Spanish and Portuguese institutions participated in this study, which was approved by the corresponding Research Ethics Board according to the Declaration of Helsinki. Informed consent was a requisite for those patients alive at the time of the data lock (May 2020).

### 2.3. Treatment Schedules

Azacitidine was administered at a standard dose of 75 mg/m^2^/d IV or SC in a 7-day schedule (days 1–7; or days 1–5, 8 and 9). Decitabine was administered at a standard dose of 20 mg/m^2^/d IV days 1–5.

### 2.4. Study Definitions and Endpoints

Cytogenetic results were classified according to the Medical Research Council (MRC) criteria [13]. AML was classified as secondary AML (sAML) as previously defined (patients with previous documented hematologic or neoplastic disease or exposure to leukemogenic agents) [14].

Estimated glomerular filtration rate (eGFR) was calculated using the Chronic Kidney Disease Epidemiology Collaboration (CKD-EPI) equation [15].

Response assessment was based on the revised recommendations of the International Working Group for Diagnosis, Standardization of Response Criteria, Treatment Outcomes and Reporting Standards for Therapeutic Trials in AML [16]. A complete remission (CR) required <5% of blast cells in bone marrow (BM) and, in peripheral blood (PB), absence of extramedullary disease, and neutrophil and platelet counts >1 × 10^9^/L and >100 × 10^9^/L, respectively. If patients did not achieve these values in PB, the response was classified as CR with incomplete recovery (CRi). Partial remission (PR) was defined as BM blasts between 5–25% and reduction ≥ 50% compared to the baseline (hematological recovery was not required). Patients with >25% blasts or reduction < 50% were assessed as resistance, and those dying before response assessment as induction death. Patients were classified as non-responders (PR or resistance) only after four–six cycles, unless they showed progression or died before. Criteria for relapse after previous CR/CRi were reappearance of disease in PB, blast cells in BM ≥ 5% or extramedullary disease.

The primary end-point was overall survival (OS) in all reported patients, comparing these results between both HMA. Secondary end-points were response to treatment, early mortality, event-free survival (EFS) and relapse-free survival (RFS), and mortality at 30, 60 and 120 days.

### 2.5. Statistical Analysis

A descriptive statistical analysis was performed after compiling the data in an Excel (Microsoft) spreadsheet. Results were expressed as percentages for categorical variables and as medians (and range) for continuous variables. Differences between the groups were evaluated by Student’s *t*-test and Mann–Whitney U-test for normally and non-normally continuous variables, respectively, and the chi-squared-test for categorical variables. CR/CRi, ORR and 120-days mortality were compared between the treatment groups using a logistic regression model.

The Kaplan–Meier estimate was used to calculate the unadjusted time-to-event variables, and the log-rank test to compare them according to the different therapeutic approaches [17,18]. The hazard ratio between the treatment groups was estimated with the Cox proportional-hazards model. OS was calculated from the date of diagnosis of AML until death in all included patients. RFS was calculated from the date of diagnosis until the date of relapse in those patients achieving CR/CRi (death without relapse was a censored event). EFS was measured from the date of diagnosis until the date of PR/resistant disease, relapse from CR/CRi or death by any cause, whichever occurred first). All P values reported are two-sided. Computations were performed using the IBM SPSS Statistics 19.0 (SPSS, Chicago, IL, USA).

## 3. Results

### 3.1. Accrual and Patient Characteristics

Between 2006 and 2019, 638 patients treated upfront with HMAs were registered. Twelve alive patients were excluded for analysis due to follow-up of less than thirty days. Thus, 626 patients were included for analysis. Of them, 487/626 (77.8%) received azacitidine and 139/626 (22.2%) received decitabine, as per physician judgement. The main characteristics are shown in Table 1. In summary, the median age was 75 years old (range, 29–89), 60% were male and median ECOG 1 (range, 0–4). Secondary AML was diagnosed in 291/614 patients (47.4%), 192/593 (32.4%) had a WBC count ≥10 × 10^9^/L and eGFR was <45 mL/min/1.73 m^2^ in 54/340 (15.9%) (Table 1). Cytogenetic risk was adverse in 204/515 patients (39.6%), and FLT3-ITD mutations or NPM1 mutations were detected in 51/269 (19%) and 33/272 (12.1%) of patients in whom molecular tests were performed.

Baseline characteristics were comparable in both groups (Table 1), except for median bone marrow blasts, which was higher in the DEC group (44%) compared with the AZA group (34%), *p* = 0.010.

### 3.2. Short-Term Outcomes

Out of the 626 patients evaluable for response, 62 (9.9%) died before the response assessment: 47/487 patients initially treated with AZA (9.7%) and 15/139 patients treated with DEC (10.8%) (*p* = 0.691). In the remaining patients, 19.1% achieved a CR (79/440 in the AZA group (17.9%) and 29/124 in the DEC group (23.4%) (*p* = 0.201)); 21.5% achieved a CR/CRi (20.5% in the AZA group vs. 25% in the DEC group (*p* = 0.276)); overall response rate (ORR = PR + CR + CRi) was 33.7% (32% in the AZA group vs. 39.5% in the DEC group (*p* = 0.120)); 59.7% resistance (60% with AZA and 58.8% with DEC); and 6.5% had stable disease/hematological improvement (7.9% with AZA and 1.6% with DEC).

A significantly lower ORR after AZA was associated with ECOG ≥2 (OR 2.98, 95% CI 1.622–5.738), secondary AML (OR 1.988, 95% CI 1.283–3.090) and estimated glomerular filtrate rate < 45 mL/min/1.73 m^2^ (OR 3.446, 95% CI 1.134–14.006), and NPM1 wild type (OR 2.204, 95% CI 1.020–4.737); while bone marrow blast count ≥ 50% was the only factor adversely influencing ORR to DEC (OR 2.317, 95% CI 1.001–5.444) (Appendix A).

The responses according to baseline characteristics are shown in Figure 1 and Figure 2. The use of DEC was associated with a significantly higher CR/CRi rate as compared to AZA in patients with ECOG ≥ 2 (OR 0.266, 95% CI 0.088–0.801), bone marrow blast count < 50% (OR 0.532, 95% CI 0.293–0.965), secondary AML (OR 0.453, 95% CI 0.223–0.918) and adverse cytogenetics (OR 0.383, 95% CI 0.171–0.857). The use of DEC was associated with a significantly higher ORR rate as compared to AZA in patients with ECOG ≥ 2 (OR 0.301, 95% CI 0.116–0.782), white blood cell count (WBC) < 10 × 10^9^/L (OR 0.543, 95% CI 0.321–0.920) and bone marrow blast count < 50% (OR 0.564, 95% CI 0.326–0.974).

Different associations were found between different variables (e.g., between the percentage of blasts in the bone marrow and the leukocyte count, or between the leukocyte count and NPM1 and FLT3-ITD mutations), but they did not influence the analysis performed for the purpose of the study.

#### 30, 60 and 120-Days Mortality

Overall, crude mortality rate at 30, 60 and 120 days after diagnosis was 5.1% (n = 32/626), 11.7% (n = 72/618) and 25.8% (n = 157/609). Death rates at 30, 60 and 120 days according to HMA were: 5.5% (n = 27/487), 11.4% (n = 55/481) and 25.4% (n = 121/476) after AZA; and 3.6% (n = 5/139), 12.4% (n = 17/137) and 27.1% (n = 36/133) after DEC, with no statistical differences between groups (*p* = 0.358, *p* = 0.754 and *p* = 0.701, respectively).

A significantly higher 120-days mortality after AZA was associated with ECOG ≥ 2 (OR 3.04, 95% CI 1.86–4.95), bone marrow blast count ≥ 50% (OR 1.91, 95% CI 1.17–3.09), high risk cytogenetic (OR 2.166, 95% CI 1.311–3.583), de novo vs. secondary AML (OR 0.62, 95% CI 0.39–0.97), adverse risk cytogenetic (OR 2.22, 95% CI 1.34–3.68), and TP53 mutated (OR 6.29, 95% CI 1.13–42.54). Age ≥ 80 years (OR 2.93, 95% CI 1.17–7.26) and WBC ≥ 10 × 10^9^/L (OR 3.43, 95% CI 1.42–8.37) were significantly associated with higher 120-days mortality after DEC, Appendix A. Non-responder patients had higher 120-days mortality after both agents (OR 8.85, 95% CI 3.48–28.70, in the AZA group, and OR 8.22, 95% CI 1.80–75.57, in the DEC group).

The use of DEC was associated with a higher 120-days mortality as compared to AZA in patients with WBC count ≥ 10 × 10^9^/L (OR 2.108, 95% CI 1.069–4.157) and those with an estimated glomerular filtrate rate ≥ 45 mL/min/1.73 m^2^ (OR 2.414, 95% CI 1.249–4.664), Figure 3.

### 3.3. Long-Term Outcomes

Regarding 1-year mortality, 336 out of 567 of the patients of the whole population with a follow-up longer than 1-year died (59.3%). There was no significant difference between AZA (58.1%) and DEC (63.6%), *p* = 0.269.

Median follow-up of patients alive was 12.29 months (range, 0.99–88.38); 12.28 months (range 0.99–88.38) in the AZA group and 12.15 (range, 0.99–47.01) in the DEC group (*p* = 0.515).

#### 3.3.1. Overall Survival

The median OS of the entire cohort was 10.0 months (CI 95%, 8.9–11.2), with 1- and 3-year OS of 40.7% and 5.3%, respectively. Median OS was 10.4 months (95% CI 9.2–11.7) for AZA vs. 8.8 (6.7–11.0) months for DEC (*p* = 0.455), Figure 4a. The 1- and 3-year OS for AZA was 41.9% and 6.1%, compared with 36.4% and 1.9% for DEC, *p* = 0.269 and *p* = 0.091, respectively. Median OS according to response was 22 months (95% CI 19.9–25.5) for CR patients, 18.8 months (95% CI 9.7–27.9) for CRi, 16.5 months (95% CI 12.4–20.5) for PR, 17.2 months (95% CI 12.6–21.8) for hematological improvement or stable disease, 7.1 months (95% CI 6.2–8.1) for resistance and 1.5 months (95% CI 0.8–1.6) for dead without hematological assessment (*p* = 0.000), Figure 4b.

Additional subgroup analyses by baseline characteristics revealed that patients ≥ 80 years (HR 1.534, 95% CI 1.005–2.341) did benefit from treatment with AZA compared with DEC, as well as patients with WBC ≥ 10 × 10^9^/L (HR 1.463, 95% CI 1.039–2.062), platelet count <20 × 10^9^/L (HR 1.984, 95% CI 1.150–3.422) and those with an estimated glomerular filtrate rate ≥ 45 mL/min/1.73 m^2^ (HR 1.464, 95% CI 1.040–2.059), Figure 5.

#### 3.3.2. Other Time-to-Event End Points

Median EFS was 5.1 months (95% CI 4.5–5.7), with no differences observed between AZA and DEC, 4.9 (95% CI 4.1–5.6) vs. 5.3 months (95% CI 4.2–6.4) (*p* = 0.824). However, patients treated with DEC had a higher median RFS than those treated with AZA (25.6 vs. 17.5 months, *p* = 0.027), Appendix A.

### 3.4. Salvage Therapy

The relapse rate of the overall population was 62.8% (76/121 patients achieving CR/CRi). The relapse rate was higher in patients treated with AZA (n = 66/90, 73.3%) compared to DEC (n = 10/31, 32.3%) (*p* < 0.0001).

Following relapse or resistance after first-line HMAs therapy, most patients received BSC only (74.5%). More patients in the DEC group received BSC (84.3% vs. 71.5%, *p* = 0.004) compared to the AZA cohort. A higher proportion of patients with adverse cytogenetics was observed among patients receiving BSC in the AZA cohort (46.6% vs. 33.7%, *p* = 0.039) than in the DEC group. No other differences were observed in this group of patients.

One hundred and thirty-five patients in the overall population received salvage treatment, 116 from the AZA cohort and 19 from the DEC cohort. Among the salvage therapies received, 26% of patients continued treatment with HMAs (35/135), with no statistically significant differences between the AZA and DEC cohorts (26% vs. 22%). In the AZA cohort, 12 patients switched to DEC (10%) while 19 continued with AZA (16%). In the DEC cohort, two patients switched to AZA (11%) and two continued with DEC (11%). Thirty eight percent of the patients (51/135) received non-intensive regimens, mainly LDAC-based, with a similar proportion among those upfront treated with AZA or DEC (37% vs. 42%). Other salvage treatments were administered in 34% of patients (33% vs. 37% among AZA or DEC cohorts, respectively), and salvage therapy was not available in the remaining patients.

Treatment response was available for 113/135 of the patients who received salvage therapy, 98 from AZA and 15 from DEC. While 13 patients achieved a CR (13.2%) and 5 (5.1%) CR/CRi in the AZA cohort, no patients in the DEC group responded to salvage therapy. In addition, 6 patients achieved a PR (6.1%), 64 were resistant (65.3%) and 8 died (8.1%) in the AZA cohort. In contrast, in the DEC cohort, 13 were resistant (86.7%) and 2 died (13.3%). Excluding those patients who died prior to response assessment, the ORR to salvage therapy was statistically significant between those upfront treated with AZA (26.7% vs. 0%, *p* = 0.035) compared to DEC.

## 4. Discussion

This study shows no significant differences in response rates or OS between upfront AZA and DEC treatment in a large retrospective with long-term follow-up cohort of AML patients. However, we identified for the first time the baseline characteristics of patients benefitting from AZA vs. DEC in terms of responses, 120-days mortality and OS. We also show differences in salvage treatment patterns and outcomes after failure to both HMAs in a real-life setting. Taken together, these findings could help to select the most appropriate HMA monotherapy regimen, or even for combination strategies.

AZA and DEC are well-tolerated and different studies suggest that they have comparable efficacy for newly diagnosed AML [5,6,10]. However, they have not been directly compared in a randomized clinical trial. In a randomized Phase 2 trial of AZA vs. DEC at low doses (3-days of each) in myelodysplastic syndromes, DEC was associated with higher response rates, with no differences in OS, but these data are not fully applicable for AML patients [19]. An indirect head-to-head comparison meta-analysis including 538 patients treated with HMAs (296 with AZA and 242 with DEC) in three randomized controlled trials showed that AZA improved OS by SUCRA analysis [8]. A recent population-based study using the Surveillance, Epidemiology, and End Results (SEER)-Medicare database showed comparable outcomes in terms of red blood cell transfusion independence and OS with the standard schedules of AZA vs. DEC [7]. However, this study lacked information about the clinical and laboratory data (including cytogenetic and molecular tests) and did not assess responses. Our study provides real-world evidence on the outcomes of AML patients treated with AZA or DEC, showing no differences in main outcomes (OS and CR/CRi), but we were able to provide detailed subgroup analyses and determine potential subsets benefiting from AZA vs. DEC.

In our study, AZA was used in a 3.5-fold proportion in contrast with other previous population studies, where the proportion was similar or even two-fold in favor of DEC [12,19]. This pattern may be explained by the delayed commercialization of DEC in Spain and Portugal and early and broader use of AZA for myelodysplastic syndromes [7,20].

The CR/CRi rate observed was 20.5% for AZA and 25% for DEC, as described in previous randomized trials. A prior Phase 3 trial of AZA achieved 27.8% of CR/CRi, while in another Phase 3 trial DEC showed a composite CR of 25.6% [5,6]. Our results also agree with other single-center and retrospective studies using HMAs that showed CR and CR/CRi of around 20% and 25% for AZA and of around 25–30% for DEC, respectively [21,22,23,24,25,26,27]. Several studies have identified prognostic factors for hematological response after AZA or DEC in AML (e.g., ECOG, WBC and cytogenetics), which were confirmed in our series [28]. Our study revealed that ECOG, eGFR and secondary AML influenced ORR in patients treated with AZA, while patients with high BM blasts percentage had an inferior ORR with DEC. Furthermore, our data suggested that using DEC could produce higher ORR as compared to AZA in patients with ECOG ≥ 2, WBC count < 10 × 10^9^/L and BM blast count < 50%.

The OS of AZA patients in this study (10.4 months) was similar to that reported in the intention-to-treat population from the AZA-AML-001 trial [6], while decitabine in our study achieved a median OS (8.8 months), slightly higher than observed in the DACO-016 trial (7.7 months) [5]. Several single center and population studies have reported median OS estimates ranging from 7.1 to 12.1 months for AZA [7,21,22,23] and 5.5 to 12.7 for DEC [7,24,25,26,27]. A recent population-based survey showed similar survival for older AML patients treated with DEC (median OS 8.2 months), while patients treated with AZA had inferior outcomes (median OS of 7.1 months). However, the DEC patients were younger and had fewer comorbid conditions than the AZA group [7]. In contrast, a significant proportion of the patients receiving AZA received a shortened 5-day schedule, which was associated with inferior outcomes. Indeed, in this study, no significant differences in survival remained after adjusting for these factors in a multivariate model [7]. More recently, a meta-analysis has shown that the AZA-treated patients exhibited lower OS when administered for five days than for a seven days schedule [10]. When only those studies using standard doses for both HMAs were included, they also found no difference in OS when comparing AZA and DEC, and the median OS observed was very similar to our study (10.83 months vs. 8.46 months, for AZA and DEC, respectively) [10]. An interesting post-hoc analysis from patients treated in the control arm of the ASTRAL-1 trial compared AZA vs. DEC and also showed no differences [9].

Our study revealed that patients ≥ 80 years, as well as those with WBC ≥ 10 × 10^9^/L, platelets < 20 × 10^9^/L, or with eGFR ≥ 45 mL/min/1.73 m^2^ did benefit from treatment with AZA as compared to DEC. The reason for these results could be explained by the lower 120-day mortality observed with AZA in these patient subgroups. However, interpretation of these findings should be tempered by the fact that the number of patients in each of these subgroups was not large, so further studies should confirm these results.

In recent years, treatment strategies for AML have evolved beyond monotherapy with HMAs for AML patients ineligible for intensive chemotherapy. The combination of venetoclax with HMAs or LDAC, or the addition of glasdegib to LDAC, has changed the front-line treatment landscape for these patients [29,30,31]. As HMAs in combination with venetoclax have evolved as a standard of care (in countries with approval), we would like to make it clear that our results should not be used to guide the decision to use AZA vs. DEC in combination with venetoclax [29,32].

There are still subgroups of AML patients who may not benefit as much from the addition of venetoclax to HMA, such as those with mutations in TP53. In our study, despite the worse impact of TP53 mutations in the 120-day mortality in the AZA group, we could not compare the impact of TP53 between AZA and DEC due to the low number of TP53-mutated patients, especially in the DEC group. To address this problem, the PETHEMA group has recently established a network of reference laboratories to provide advanced molecular diagnostics for AML patients in the context of our scientific working group [33]. Ongoing and future analyses will allow the evaluation of the efficacy of AZA vs. DEC, according to detailed genomic characteristics. For these high-risk patients with mutated TP53, promising drugs are being developed to restore its function as a tumor suppressor gene [34,35].

Finally, we described salvage therapy received after relapse or resistance in these patients. To our knowledge, there are no studies evaluating and comparing salvage therapy after first-line treatment with HMAs. It is striking that, although salvage treatments were similar in both treatment groups, the response rate was very different between patients treated at first line with AZA and DEC, which could explain the similar OS between AZA and DEC despite the higher RFS in the DEC group. This could help us to choose the first line with successive lines in mind.

## 5. Conclusions

In conclusion, we found no differences in the response rates and OS between first-line treatment with AZA or DEC in a large retrospective cohort of newly-diagnosed AML patients with long-term follow-up. Beyond these data, our study reveals that there may be subgroups that appear to have reacted more positively with AZA or DEC. These new findings might be helpful in the selection of the more appropriate HMA in monotherapy, but it is difficult to interpret these findings in a non-randomized retrospective study.

## Figures and Tables

**Figure 1 cancers-14-02342-f001:**
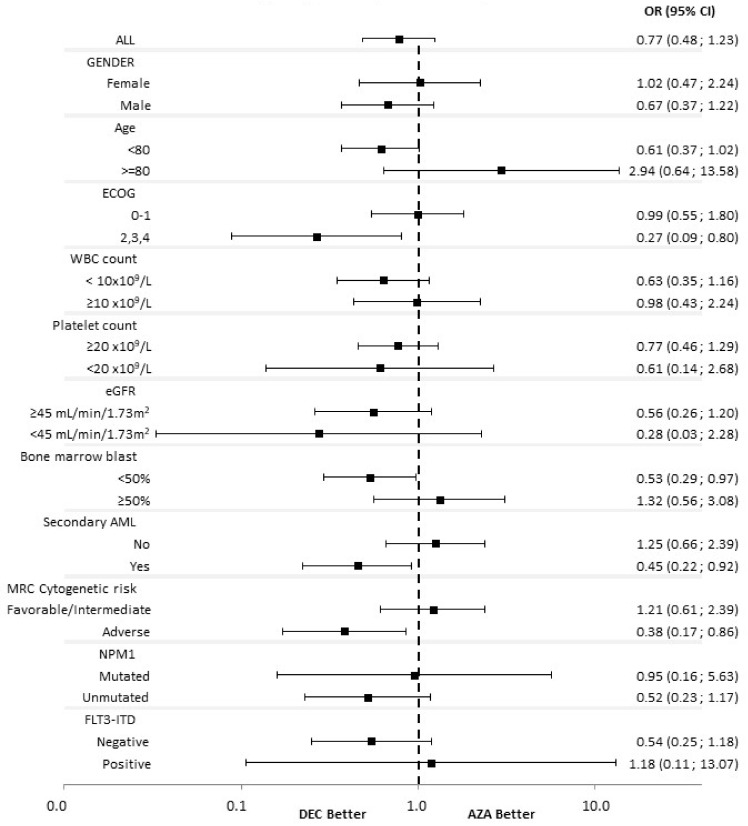
Subgroup analysis for CR/CRi. AML, Acute myeloid leukemia; AZA, azacitidine; CR, complete remission; CRi, complete remission with incomplete blood count recovery; DEC, decitabine; ECOG, Eastern Cooperative Oncology Group; eGFR, estimated glomerular filtrate rate; FLT3, FMS-like tyrosine kinase 3; ITD: internal tandem duplication; MRC, Medical Research Council; NPM1, Nucleophosmin1; WBC, white blood cells.

**Figure 2 cancers-14-02342-f002:**
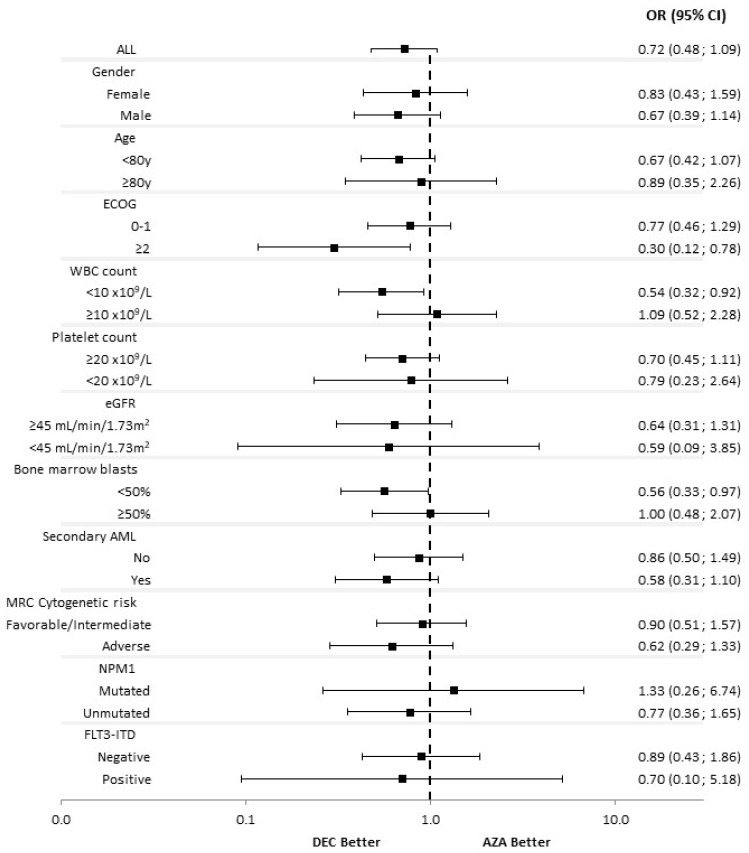
Subgroup analysis for ORR. AML, Acute myeloid leukemia; AZA, azacitidine; CR, complete remission; CRi, complete remission with incomplete blood count recovery; DEC, decitabine; ECOG, Eastern Cooperative Oncology Group; eGFR, estimated glomerular filtrate rate; FLT3, FMS-like tyrosine kinase 3; ITD: internal tandem duplication; MRC, Medical Research Council; NPM1, Nucleophosmin1; ORR, overall response rate (CR + CRi + PR); PR, partial remission; WBC, white blood cells.

**Figure 3 cancers-14-02342-f003:**
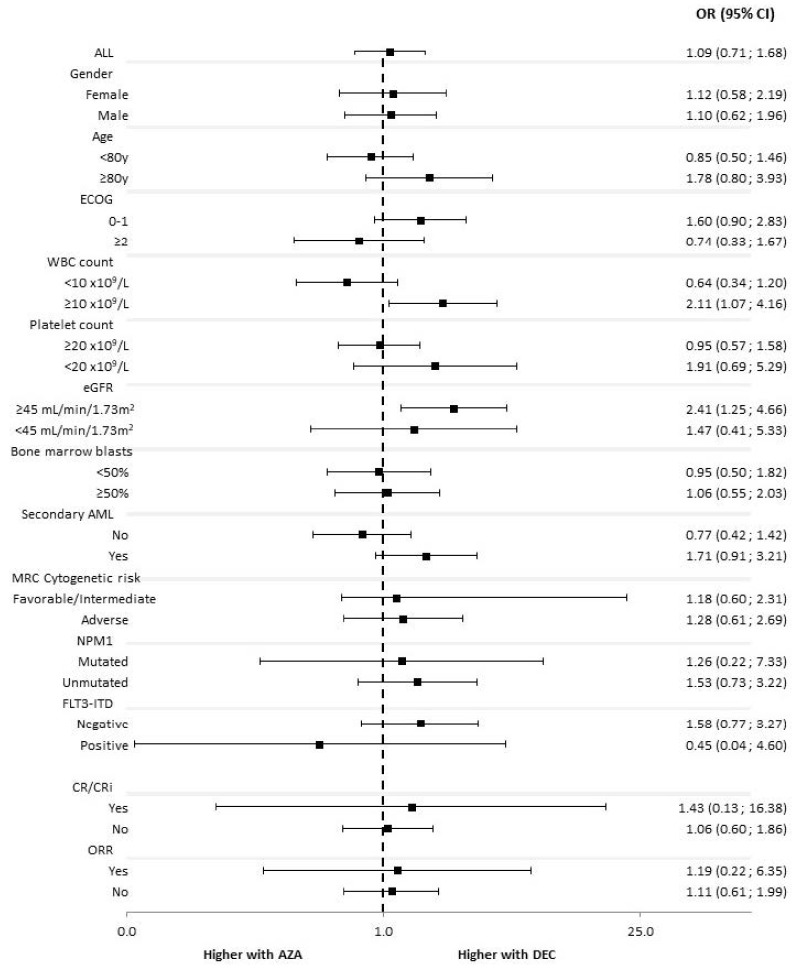
Subgroup analysis for 120-day mortality. AML, Acute myeloid leukemia; AZA, azacitidine; CR, complete remission; CRi, complete remission with incomplete blood count recovery; DEC, decitabine; ECOG, Eastern Cooperative Oncology Group; eGFR, estimated glomerular filtrate rate; FLT3, FMS-like tyrosine kinase 3; ITD: internal tandem duplication; MRC, Medical Research Council; NPM1, Nucleophosmin1; ORR, overall response rate (CR + CRi + PR); PR, partial remission; WBC, white blood cells.

**Figure 4 cancers-14-02342-f004:**
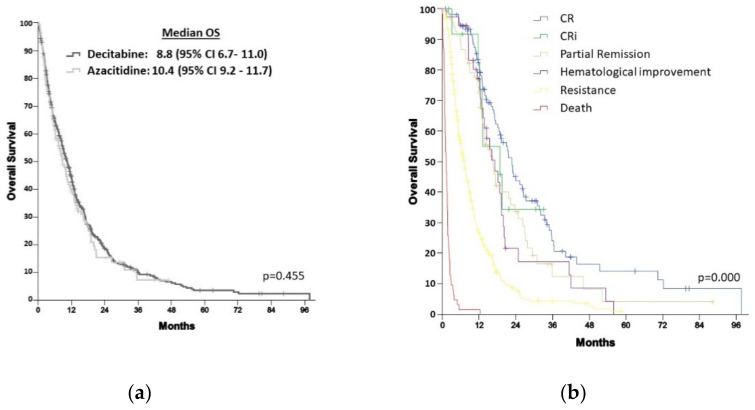
(**a**) Overall survival among patients treated with azacitidine vs. decitabine; (**b**) Overall survival according to response. CR, complete remission; CRi, complete remission with incomplete blood count recovery; OS, overall survival; PR, partial remission.

**Figure 5 cancers-14-02342-f005:**
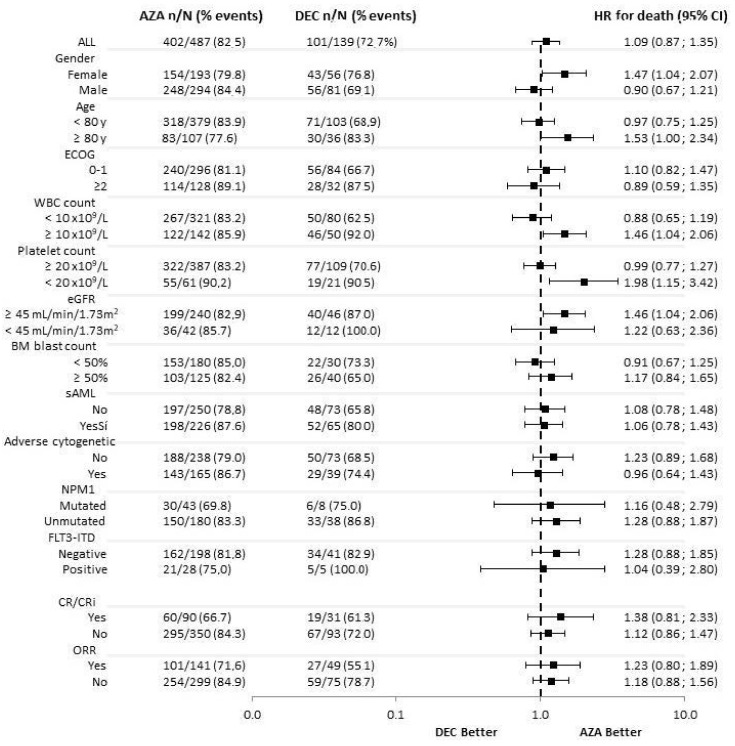
Subgroup Analysis of Overall Survival. AML, Acute myeloid leukemia; AZA, azacitidine; BM, bone marrow; CR, complete remission; CRi, complete remission with incomplete blood count recovery; DEC, decitabine; ECOG, Eastern Cooperative Oncology Group; eGFR, estimated glomerular filtrate rate; FLT3, FMS-like tyrosine kinase 3; ITD: internal tandem duplication; NPM1, Nucleophosmin1; ORR, overall response rate (CR + CRi + PR); PR, partial remission; sAML, secondary AML; WBC, white blood cells.

**Table 1 cancers-14-02342-t001:** Baseline characteristics of the study population.

Baseline Characteristics	Overall		Azacitidine		Decitabine		*p*-Value
	Median (Range)	n (%)	Median (Range)	n (%)	Median (Range)	n (%)	
Total		626 (100)		487 (100)		139 (100)	
Gender (male)		375 (60.1)		294 (60.4)		81 (59.1)	0.793
Age	75.06 (29.18–89.85)	625 (100)	74.89 (29.18–88.56)	486 (100)	75.68 (53.76–89.85)	139 (100)	0.085
<70		144 (23.0)		121 (24.9)		23 (16.5)	0.205
70–74		166 (26.6)		125 (25.7)		41 (29.5)	
75–79		172 (27.5)		133 (27.4)		39 (28.1)	
≥80		143 (22.9)		107 (22.0)		36 (25.9)	
Secondary AML		291 (47.4)		226 (47.5)		65 (47.1)	0.938
AML-t		46 (19.9)		38 (19.6)		8 (21.6)	0.776
ECOG performance status							
0		115 (21.3)		92 (21.7)		23 (19.8)	
1		265 (49.1)		204 (48.1)		61 (52.6)	
2		122 (22.6)		99 (23.3)		23 (19.8)	
3		36 (6.7)		28 (6.6)		8 (6.9)	
4		2 (0.4)		1 (0.2)		1 (0.9)	
ECOG 0–1		382 (70.5)		296 (69.8)		84 (72.4)	0.586
ECOG ≥ 2		160 (29.5)		128 (30.2)		32 (27.6)	
WBC count (×10^9^/L)		4.0 (0.26–214.0)		3.9 (0.26–174.31)		5.09 (0.50–214.0)	0.081
<10 × 10^9^/L		401 (67.6)		321 (69.3)		80 (61.5)	0.093
≥10 × 10^9^/L		192 (32.4)		142 (30.7)		50 (38.5)	
Hemoglobin (g/dL)	8.9 (2.00–14.90)	574 (100)	8.9 (2.00–14.90)	446 (100)	9 (4.70–14.90)	128 (100)	0.123
Platelet count (×10^9^/L)	58.85 (3.0–1140.0)	578 (100)	60 (6.0–1140.0)	448 (100)	56 (3.0–822.0)	130 (100)	0.854
Platelet ≥ 20 × 10^9^/L Platelet < 20 × 10^9^/L		496 (85.8) 82 (14.2)		387 (86.4) 61 (13.6)		109 (83.8) 21 (16.2)	0.465
Creatinine (mg/dL)	0.93 (0.40–7.00)	340 (100)	0.9 (0.40–7.00)	282 (100)	1.02 (0.48–5.30)	58 (100)	0.141
eGFR ≥ 45 mL/min/1.73 m^2^		286 (84.1)		240 (85.1)		46 (79.3)	0.271
eGFR < 45 mL/min/1.73 m^2^		54 (15.9)		42 (14.9)		12 (20.7)	
LDH (U/L)	387 (26–5168)	409	382 (26–5168)	348	422 (153–3027)	61	0.085
Albumin (g/dL)	3.8 (2.20–6.20)	269	3.78 (2.20–6.20)	222	3.86 (2.70–5.20)	47	0.776
Bone marrow blast count, %	36 (0–100)	565	34 (0–96)	436	44 (1–100)	129	0.010
<30%		210 (37.2)		180 (41.3)		30 (23.3)	0.000
≥30% to <50%		165 (29.2)		125 (28.7)		40 (31.0)	
≥50%		190 (33.6)		131 (30.0)		59 (45.7)	
FAB subtype		407 (100)		349 (100)		58 (100)	0.853
M0/M6/M7		56 (13.8)		49 (14.0)		7 (12.1)	
M1/M2		101 (24.8)		84 (24.1)		17 (29.3)	
M4/M5		72 (17.7)		62 (17.8)		10 (17.2)	
Other		178 (43.7)		154 (44.1)		24 (41.4)	
MRC Cytogenetic risk		515 (100)		403 (100)		112 (100)	0.241
Favorable/Intermediate		311 (60.4)		238 (59.1)		73 (65.2)	
Adverse		204 (39.6)		165 (40.9)		39 (35.8)	
Somatic mutations							
NPM1		52/269 (19.0)		43/223 (19.3)		8/46 (17.4)	0.766
FLT3-ITD		33/272 (12.1)		28/226 (12.4)		5/46 (10.9)	0.774
TP53		13/90 (14.4)		11/71 (15.5)		2/19 (10.5)	0.728

AML—Acute myeloid leukemia; AML-t—acute myeloid leukemia related with therapy; ECOG—Eastern Cooperative Oncology Group; eGFR—estimated glomerular filtrate rate; FAB—French-American-British; FLT3—FMS-like tyrosine kinase 3; ITD—internal tandem duplication; LDH—lactate dehydrogenase; MRC—Medical Research Council; MRC—Myelo; NPM1—Nucleophosmin1; WBC—white blood cells.

## Data Availability

Data are property of the PETHEMA foundation. Data may be available from the corresponding author, upon reasonable request.

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
