# Peer review of "Azacitidine vs. Decitabine in Unfit Newly Diagnosed Acute Myeloid Leukemia Patients: Results from the PETHEMA Registry"

_cancers, 2022, doi:10.3390/cancers14092342_

Round 1
Reviewer 1 Report
Labrador et al. present a well-written analysis of responses and survival outcomes of AML patients treated with AZA and DEC in the PETHEMA registry and identify patient and disease characteristics associated with differences in response rates. This is a methodologically sound and interesting study although the implications for practice are limited as HMA monotherapy is no longer the standard of care for most patients with AML who are ineligible for intensive chemotherapy. Additionally, I have the following minor comments:
- Line 90-93: The statement that DEC had a "significantly higher median OS" compared to BSC or LDAC in DACO-016 is misleading. There was no difference in median OS at the primary analysis but only in an unplanned subsequent analysis. This needs to be corrected. Similarly, the OS benefit with AZA in the study by Dombret et al. was not significant for the intention-to-treat population
- Figure 1: Figure 1 uses "," to indicate decimals, while "." is used in the text. Please change this in the figure for consistency. This applies to all figures.
- Short-term outcomes/response rates: The authors identify several disease and patient characteristics associated with a higher response rate to DEC. Was this a univariate analysis or a multivariate logistic regression analysis (if I am understanding the methods correctly the latter was used)? Can the authors analyze whether some of these variables (e.g., adverse cytogenetics, secondary AML, low blast count) are correlated?
- While there has been no direct head-to-head comparison in a randomized trial yet, an interesting post-hoc analysis from patients treated in the control arm of the ASTRAL-1 trial compared AZA vs DEC and also showed no differences. This might be worthwhile to include (Paper: Comparative Results of Azacitidine and Decitabine from a Large Prospective Phase 3 Study in Treatment Naive Patients with Acute Myeloid Leukemia Not Eligible for Intensive Chemotherapy (confex.com))
- Given how rapidly the frontline treatment landscape of AML patients ineligible for intensive chemotherapy is changing, I am not sure how applicable these data to current practice are and do not think the data presented here can be used to guide selection of AZA vs DEC as a combination partner with, for example, venetoclax. Whether AZA/VEN is different than DEC/VEN is unclear and the last sentence of the conclusion should be revised.
Author Response
Reviewer #1:
Labrador et al. present a well-written analysis of responses and survival outcomes of AML patients treated with AZA and DEC in the PETHEMA registry and identify patient and disease characteristics associated with differences in response rates. This is a methodologically sound and interesting study although the implications for practice are limited as HMA monotherapy is no longer the standard of care for most patients with AML who are ineligible for intensive chemotherapy.
Additionally, I have the following minor comments:
Line 90-93: The statement that DEC had a "significantly higher median OS" compared to BSC or LDAC in DACO-016 is misleading. There was no difference in median OS at the primary analysis but only in an unplanned subsequent analysis. This needs to be corrected. Similarly, the OS benefit with AZA in the study by Dombret et al. was not significant for the intention to-treat population.
Response: Thank you for your comment. We agreed with the reviewer and rephrased the paragraph: “Both HMAs have been associated with a clinically meaningful improvement in overall survival (OS) compared to LDAC, although the primary end point was not met [5,6]. In the phase 3 DACO-016 trial, DEC achieved a significantly higher median OS (7.7 months, 95%CI: 6.2 - 9.2) compared to patients receiving therapy choice, considered best supportive care (BSC) or LDAC (5.0 months, 95%CI: 4.3 - 6.3) in an ad hoc mature analysis in the intent-to-treat population [5]. Similarly, in the phase 3 AML-001 trial, AZA patients exhibited a median OS (10.4 months, 95%CI: 8.0 - 12.7) compared to conventional care regimens, considered standard IC, LDAC or BSC (6.5 months, 95%CI: 5.0 - 8.6) [6]”.
Figure 1: Figure 1 uses "," to indicate decimals, while "." is used in the text. Please change this in the figure for consistency. This applies to all figures.
Response: Thank you for your comment. We have changed all figures for consistency.
Short-term outcomes/response rates: The authors identify several disease and patient characteristics associated with a higher response rate to DEC. Was this a univariate analysis or a multivariate logistic regression analysis (if I am understanding the methods correctly the latter was used)? Can the authors analyze whether some of these variables (e.g., adverse cytogenetics, secondary AML, low blast count) are correlated?
Response: Thank you for your comment. A logistic regression analysis was performed including each variable separately along with AZA or DEC treatment. Different associations were found between different variables (e.g. between the percentage of blasts in the bone marrow and the leukocyte count, or between the leukocyte count and NPM1 and FLT3-ITD mutations), but they did not influence the analysis performed for the purpose of the study.
While there has been no direct head-to-head comparison in a randomized trial yet, an interesting post-hoc analysis from patients treated in the control arm of the ASTRAL-1 trial compared AZA vs DEC and also showed no differences. This might be worthwhile to include (Paper: Comparative Results of Azacitidine and Decitabine from a Large Prospective Phase 3 Study in Treatment Naive Patients with Acute Myeloid Leukemia Not Eligible for Intensive Chemotherapy (confex.com))
Response: Thank you for your suggestion. We did not know this interesting analysis, so we have included in our paper (reference 9).
Given how rapidly the frontline treatment landscape of AML patients ineligible for intensive chemotherapy is changing, I am not sure how applicable these data to current practice are and do not think the data presented here can be used to guide selection of AZA vs DEC as a combination partner with, for example, venetoclax. Whether AZA/VEN is different than DEC/VEN is unclear and the last sentence of the conclusion should be revised
Response: Thank you very much for your comment. You are indeed right. We have omitted part of the last sentence of the conclusion that referred to the possibility of choosing the best HMA for the combination study (e.g. with venetoclax): “These new findings might be helpful in the selection of the more appropriate HMA in monotherapy”. We have also modified it in the simple summary.
In addition, we have included this paragraph in the discussion: “In recent years, treatment strategies for AML have evolved beyond monotherapy with HMAs for AML patients ineligible for intensive chemotherapy. The combination of venetoclax with HMAs or LDAC, or the addition of glasdegib to LDAC, has changed the front-line treatment landscape for these patients [29–31]. As HMAs in combination with venetoclax have evolved as a standard of care (in countries with approval), we would like to make it clear that our results should not be used to guide the decision to use AZA versus DEC in combination with venetoclax [29, 32].”
We hope that the Editor and the Reviewers will find our modified manuscript suitable for publication in Cancers.
We look forward to hearing from you at your earliest convenience.
Thank you very much for all your kindness.
Sincerely yours,
Jorge Labrador
Hematology Department
Research Unit
Hospital Universitario de Burgos, Burgos, Spain
jlabradorg@saludcastillayleon.es

Reviewer 2 Report
This is a very interesting article. I think the main takeaway is that in this very large registry study, overall azacitidine and decitabine perform similarly. There may be some subgroup differences, but in general outcomes seemed comparable, which is important to convey to the AML field. I have two major comments and several minor revisions but overall enjoyed reading the paper.
Major revision(s):
1) Hypomethylating agent (HMA) monotherapy has largely been replaced by the doublet of HMA + venetoclax based on the results of the VIALE-A trial and several smaller single arm studies showing the efficacy of HMA+venetoclax. Thus, it would be nice if the authors stated more in the discussion how their findings may (or may not, which is also ok) influence care in the current landscape of HMA+venetoclax. They could discuss in more depth if their should be preference for AZA + venetoclax (as in VIALE-A) or for DEC + venetoclax (as described in DiNardo et al., Lancet Haematology 2020). There are still subgroups of AML that may not benefit as much from the addition of venetoclax to HMA, such as those with mutations in TP53, FLT3 (ITD), and RAS, and thus these may be groups where the authors could discuss the role of HMA monotherapy (and AZA vs DEC) or about which HMA to choose as a backbone for clinical trials testing HMA + targeted therapy for AML with these mutations.
2) I would be careful of overstating the conclusion from this paper that there are subgroups who benefit from AZA or DEC. For example, it does not clearly make sense why patients with better renal function would benefit from AZA compared to DEC. Particularly in a retrospective study (not a randomized study), trying to read too much into subgroup analyses is risky as you may find associations that may not be reproducible and are, at best, hypothesis-generating. For instance, the eGFR >45 is almost crossing 1.0 in the 95% CI in Figure 5. I think the data in this study is strong enough as is, even without stating which subgroups might benefit. I defer to the authors, but it is ok to also just display the results and state that overall outcomes seemed similar (which is quite important itself in such a large registry), and that there might be subgroups who seem to have done better with AZA or DEC but it is hard to interpret these findings in a non-randomized retrospective study.
Minor revisions:
1) review of small spelling/grammatical changes, such as
- line 62: 120-day (not days) mortality
- line 71 and 71: CR was (not were) 18% with AZA and CR/CRi was (not were) 20.5%
- line 98: "To better select the HMA agent" could be reworded
- line 141: not sure if "toxic death" is the way this should be phrased
- line 155 and 156: references are within the sentence here but end of sentence in rest of paper
2) line 159: RFS was calculated from date of diagnosis; this is ok because the definition used in this study is clearly stated. However in the AML ELN 2017 guidelines, RFS is defined from date of CR/CRi till relapse or death
3) Table 1: overall column, age: % do not add up to 100%, looks like 70-74yo has an error next to 166
4) in the paper, please change P53 to TP53 as that is the nomenclature for the gene typically used and p53 for the protein
5) consider changing Supplementary Figure 1 to state "lack of response" rather than No ORR
6) line 311; it is true that there are no randomized studies of AZA vs DEC in AML; however, there was a randomized phase 2 trial of AZA vs DEC in MDS, albeit with 3d of each rather than 7d and 5d (respectively); I defer to the authors, but could consider a brief reference to this if indicated (Jabbour et al., Blood 2017)
Author Response
Reviewer #2:
This is a very interesting article. I think the main takeaway is that in this very large registry study, overall azacitidine and decitabine perform similarly. There may be some subgroup differences, but in general outcomes seemed comparable, which is important to convey to the AML field. I have two major comments and several minor revisions but overall enjoyed reading the paper.
Major revision(s):
1) Hypomethylating agent (HMA) monotherapy has largely been replaced by the doublet of HMA + venetoclax based on the results of the VIALE-A trial and several smaller single arm studies showing the efficacy of HMA+venetoclax. Thus, it would be nice if the authors stated more in the discussion how their findings may (or may not, which is also ok) influence care in the current landscape of HMA+venetoclax. They could discuss in more depth if their should be preference for AZA + venetoclax (as in VIALE-A) or for DEC + venetoclax (as described in DiNardo et al., Lancet Haematology 2020). There are still subgroups of AML that may not benefit as much from the addition of venetoclax to HMA, such as those with mutations in TP53, FLT3 (ITD), and RAS, and thus these may be groups where the authors could discuss the role of HMA monotherapy (and AZA vs DEC) or about which HMA to choose as a backbone for clinical trials testing HMA + targeted therapy for AML with these mutations.
Response: Thank you very much for your comment. We have included these paragraphs in the discussion:
“In recent years, treatment strategies for AML have evolved beyond monotherapy with HMAs for AML patients ineligible for intensive chemotherapy. The combination of venetoclax with HMAs or LDAC, or the addition of glasdegib to LDAC, has changed the front-line treatment landscape for these patients [29–31]. As HMAs in combination with venetoclax have evolved as a standard of care (in countries with approval), we would like to make it clear that our results should not be used to guide the decision to use AZA versus DEC in combination with venetoclax [29,32].”
Regarding those patients with TP53 mutations, unfortunately, “we could not compare the impact of TP53 between AZA and DEC due to low number of TP53 mutated patients, especially in DEC group. To address this problem, the PETHEMA group has recently established a network of reference laboratories to provide advanced molecular diagnostics for AML patients in the context of our scientific working group. Ongoing and future analyses will allow evaluating the efficacy of AZA vs. DEC according to detailed genomic characteristics. For these high-risk patients with mutated TP53, promising drugs are being developed to restore its function as a tumor suppressor gene[34,35].”
2) I would be careful of overstating the conclusion from this paper that there are subgroups who benefit from AZA or DEC. For example, it does not clearly make sense why patients with better renal function would benefit from AZA compared to DEC. Particularly in a retrospective study (not a randomized study), trying to read too much into subgroup analyses is risky as you may find associations that may not be reproducible and are, at best, hypothesis-generating. For instance, the eGFR >45 is almost crossing 1.0 in the 95% CI in Figure 5. I think the data in this study is strong enough as is, even without stating which subgroups might benefit. I defer to the authors, but it is ok to also just display the results and state that overall outcomes seemed similar (which is quite important itself in such a large registry), and that there might be subgroups who seem to have done better with AZA or DEC but it is hard to interpret these findings in a non-randomized retrospective study.
Response: Thank you very much for your comment. We have modified the conclusions in line with your comment:
“In conclusion, we found no differences in response rates and OS between first-line treatment with AZA or DEC in a large retrospective cohort of newly-diagnosed AML patients with long-term follow-up. Beyond these data, our study reveals that there may be subgroups that appear to have done better with AZA or DEC. These new findings might be helpful in the selection of the more appropriate HMA in monotherapy, but it is difficult to interpret these findings in a non-randomised retrospective study.”
Minor revisions:
1) review of small spelling/grammatical changes, such as
line 62: 120-day (not days) mortality
line 71 and 71: CR was (not were) 18% with AZA and CR/CRi was (not were) 20.5%
line 98: "To better select the HMA agent" could be reworded
line 141: not sure if "toxic death" is the way this should be phrased
line 155 and 156: references are within the sentence here but end of sentence in rest of paper
Response: Thank you very much for your comment. We have we have corrected all the grammatical errors noted.
2) line 159: RFS was calculated from date of diagnosis; this is ok because the definition used in this study is clearly stated. However in the AML ELN 2017 guidelines, RFS is defined from date of CR/CRi till relapse or death
Response: We appreciate the reviewer's comment and will take it into account for future studies.
3) Table 1: overall column, age: % do not add up to 100%, looks like 70-74yo has an error next to 166
Response: Thank you very much for the correction, we apologise for the typographical error. We have corrected table 1 accordingly.
4) in the paper, please change P53 to TP53 as that is the nomenclature for the gene typically used and p53 for the protein
Response: Thank you for the correction. We have corrected it according to the established nomenclature.
5) consider changing Supplementary Figure 1 to state "lack of response" rather than No ORR.
Response: Thank you for the correction. We have changed Supplementary Figure 1 according to the comments of reviewers 1 and 2.
6) line 311; it is true that there are no randomized studies of AZA vs DEC in AML; however, there was a randomized phase 2 trial of AZA vs DEC in MDS, albeit with 3d of each rather than 7d and 5d (respectively); I defer to the authors, but could consider a brief reference to this if indicated (Jabbour et al., Blood 2017)
Response: Thank you for your comment. We have included the following sentence in the discussion: “AZA and DEC are well-tolerated and different studies suggest that they have comparable efficacy for newly diagnosed AML [5,6,10]. However, they have not been directly compared in a randomized clinical trial. In a randomized phase 2 trial of AZA vs DEC at low doses (3-day of each) in myelodysplastic syndromes, DEC was associated with higher response rates, with no differences in OS, but these data are not fully applicable for AML patients [19].”
We hope that the Editor and the Reviewers will find our modified manuscript suitable for publication in Cancers.
We look forward to hearing from you at your earliest convenience.
Thank you very much for all your kindness.
Sincerely yours,
Jorge Labrador
Hematology Department
Research Unit
Hospital Universitario de Burgos, Burgos, Spain
jlabradorg@saludcastillayleon.es
